# Serum Caspase-3 Levels as a Predictive Molecular Biomarker for Acute Ischemic Stroke

**DOI:** 10.3390/ijms25126772

**Published:** 2024-06-20

**Authors:** Andrei-Lucian Zaharia, Violeta Diana Oprea, Camelia Alexandra Coadă, Dana Tutunaru, Aurelia Romila, Bianca Stan, Ana Croitoru, Ana-Maria Ionescu, Mihaiela Lungu

**Affiliations:** 1Faculty of Medicine and Pharmacy, “Dunarea de Jos” University of Galati, 800216 Galati, Romania; zaharia.andreilucian@gmail.com (A.-L.Z.); diana.v.oprea@gmail.com (V.D.O.); aurelia.romila@yahoo.com (A.R.); csb.bianca@gmail.com (B.S.); croitoruana28@yahoo.com (A.C.); mihaelalungu17@yahoo.com (M.L.); 2“St. Apostle Andrei” Clinical Emergency County Hospital Galati, 800578 Galati, Romania; 3Faculty of Medicine, “Iuliu Haţieganu” University of Medicine and Pharmacy, 400012 Cluj-Napoca, Romania; 4Faculty of Medicine and Pharmacy, Ovidius University of Constanța, 900470 Constanța, Romania; iuliusana@gmail.com

**Keywords:** caspase-3, biomarker, acute ischemic stroke, early diagnosis

## Abstract

Caspases are key players in the apoptotic process and have been found to contribute to the pathogenesis of a variety of diseases, including neurological disorders such as ischemic stroke. This study aimed to investigate the serum levels of Caspase-3 in patients with acute ischemic stroke (AIS) and in control patients without ischemic events. Moreover, we explored any potential associations with the clinical outcomes of AIS. We enrolled 69 consecutive patients with clinical signs and symptoms of AIS in the presence of a negative CT scan who presented themselves at the Clinical Neurological Department from the Emergency Clinical Hospital of Galati within the first 24 h of symptom onset. The control group comprised 68 patients without cerebral ischemic pathologies. A comparison of the two groups showed significantly higher levels of caspase-3 at 24 and 48 h after hospital admission. No significant associations between caspase-3 levels and clinical features of AIS were seen. However, in a subgroup analysis conducted on patients with moderate/severe and severe stroke, lower levels of caspase-3 were associated with early mortality. Caspase-3 levels did not directly correlate with AIS severity or prognosis when considering all AIS patients. In patients with moderate to severe National Institute of Health Stroke Scale (NIHSS) scores, caspase-3 might be a prognostic indicator of early death. Further studies are required to confirm these results and further explore the mechanisms behind these findings.

## 1. Introduction

While stroke is considered the second largest cause of mortality worldwide, acute ischemic stroke (AIS) accounts for approximately 85% of all strokes, killing yearly more than 2.7 million people globally [1]. In AIS, vessel occlusion leads to regional brain hypoperfusion, usually resulting in the irreversible death of brain tissue if there is no prompt restoration of blood flow. Early diagnosis is therefore needed to allow fast interventions, and the established standards currently require clinical and imagistic assessments. The clinical assessment of stroke severity uses the National Institute of Health Stroke Scale (NIHSS), which adds to the gold-standard approach of evaluation based on brain and neurovascular imaging. Yet many cases present a scarce clinical and/or imagistic picture, which delays the confirmation of stroke suspicion. Furthermore, computed tomography (CT) techniques may have a low sensitivity in the early hours of onset, particularly in the case of minor vertebrobasilar strokes. On the other hand, magnetic resonance imaging (MRI) has higher sensitivity, but it is not as widely and readily available as CT scans, thus screening patients by MRI is out of reach if not impossible for small centers [1,2].

Having additional blood biomarkers could represent an important and valuable diagnostic tool for the clinician, especially in the early stages of AIS. The further ability to quickly differentiate between ischemic and hemorrhagic stroke, combined with the specificity for cerebral processes, would be the ideal characteristics of a valuable AIS biomarker. For example, brain-derived molecules that are transferred to the bloodstream due to the destruction of the blood–brain barrier (BBB) occurring during stroke might present these characteristics.

Recent studies provided data on the potential diagnostic value of using one or a combination panel of biomarkers such as glial proteins (like GFAP—glial fibrillary acidic protein), neuron-specific enolase (NSE), matrix metalloproteinase 9 (MMP-9), brain natriuretic peptide (BNP), N-methyl-D-aspartate receptor proteins, or apolipoproteins. For example, a study showed that interleukin-6 (IL-6) expression was upregulated following brain ischemia when measured in the first 24 h in the peripheral blood [3]. Sotgiu et al. [4] presented evidence that tumor necrosis factor-α (TNF-α) reaches a high concentration in the first 6 h after AIS and it could be strongly correlated with the clinical severity and extent of the brain infarct. Nevertheless, most inflammatory biomarkers lack specificity and have limited standardization, making the diagnosis of AIS challenging in paucisymptomatic patients.

Necrosis and apoptosis are two major mechanisms of cell death in the central nervous system (CNS) in brain injuries including stroke, and it was proven that during apoptosis, cell death is caused by a tightly regulated biochemical cascade involving activation of caspases (cysteine-aspartic-proteases) [5,6]. The caspase family currently has 14 known members. The involvement of caspases at the level of the CNS is extremely complex, as they participate in multiple pathways such as inflammation, metabolism, and autophagy. The caspase family has been shown to play multiple roles in processes involving aging, inflammation, pyroptosis, genomic stability maintenance, and homeostasis of the adult T-cell population, as well as in brain injury progression [7,8,9,10,11].

In conditions of stroke/cerebral ischemia, increased activity of caspase-9 causes loss of the highly differentiated cells, causing functional decline and reduced regenerative ability. Caspase-9 deficiency leads to misrouted axons, impaired synapse formation, and defects in sensory neurons [7], especially those responsible for olfaction. Sugawara et al. demonstrated that cleaved caspase-9 was increased in the brain 12 h after ischemia with an important role in neuronal death [12].

Activated caspase-3 upregulation at acute time points after experimental traumatic brain injuries was observed primarily in neurons and to a lesser extent, in astrocytes and oligodendrocytes [13]. Expression of cleaved caspase-8 and caspase-3 have been reported in Iba1-positive microglia in animal models of cerebral ischemia and in CD68-positive microglia/macrophages in ischemic human stroke. Rosell et al. [14] provided evidence that plasma caspase-3 levels were higher in stroke patients versus the control group throughout the acute phase of stroke. At 24 h after stroke onset, these levels were associated with poorer short- and long-term neurological outcomes and positively correlated with the infarct volume.

The limited number of clinical studies focusing on caspase-3 as an early stroke biomarker, as well as the lack of consistency throughout the results supporting its clinical relevance for AIS diagnosis, triggered our interest in the research of this particular molecule. Our prospective study started from the evidence of caspase-3 role in brain tissue damage induced by cerebral ischemia and aimed to determine if there is any correlation between serum caspase-3 levels and AIS features, as well as if there are any relevant differences between AIS patients and control cases without ischemic events.

## 2. Results

### 2.1. Description of the Study Population

A total of 69 patients with AIS were included in the study, while the control group comprised 68 subjects. The AIS group comprised 26 (37.68%) patients with cardioembolic stroke and 43 (62.32%) with atherothrombotic stroke. Most patients (31, 44.93%) had left sylvian AIS, followed by 24 (34.78%) patients with right sylvian AIS, and 14 (20.29%) with the vertebrobasilar system affected. In terms of severity, 22 (31.88%) patients had a minor stroke, 27 (39.13%) moderate stroke, 15 (21.74%) moderate-to-severe stroke, and only 5 (7.25%) patients had a severe stroke.

General characteristics of the patients can be found in Table 1. No significant differences were seen in terms of demographic data regarding gender, age, and residential background. Regarding comorbidities, diabetes and grade 3 hypertension were more frequently seen in AIS patients than in controls (*p* = 0.013 and 0.033, respectively) (Table 1).

### 2.2. Caspase-3 Serum Levels in AIS and Control Subjects

Measurement of caspase-3 at patient presentation showed significantly higher levels in AIS patients than in control subjects (controls: 1.13 (0.53; 1.91) vs. AIS patients: 5.1 (3.4; 7.9); *p* < 0.001) (Figure 1A). The power calculation for this analysis was 98.02%. Moreover, repeated measures in AIS patients performed 48 h after hospitalization showed some variation between timepoints with some patients presenting increasing levels while others showing decreasing levels. Nevertheless, caspase-3 levels remained constant in the patients’ blood (T1: 5.10(3.42; 7.90) vs. T2: 5.35(3.45; 8.03), *p* = 0.899) (Figure 1B).

Logistic regression analysis revealed that caspase-3 correctly classified 84.7% of the patients. In detail, patients had a hazard ratio of 2.08 of receiving an AIS diagnosis (HR = 2.08 95% CI = 1.65; 2.75 *p* < 0.001). The best threshold of serum caspase-3 for discrimination between AIS and controls was 2.50 ng/mL.

### 2.3. Caspase-3 Serum Levels and Features of AIS Patients

Next, we sought to investigate whether caspase-3 levels were associated with clinical features and outcomes of the AIS patients. Analysis of health conditions and AIS-specific characteristics showed an absence of any discernible correlation with caspase-3 levels. These include common comorbidities such as atrial fibrillation, blood hypertension, dyslipidemia, obesity, and diabetes, as well as AIS type, vascular territory, NIHSS, and early death (Table 2).

We further evaluated whether the clinical and paraclinical investigations completed at the patient’s admission varied with caspase-3 serum levels and found a slight positive association between caspase-3 and hemoglobin levels (r = 0.29; *p* < 0.05) (Figure 2). As expected, NIHSS was positively associated with age and AST levels.

Considering that the NIHSS score is the most important predictor of mortality among AIS patients [15,16], we sought to investigate whether additional risk stratifications could be useful in the case of patients presenting with NIHSS scores above 16, corresponding to the severity classes of moderate/severe and severe. Of all the AIS patients analyzed, a total of 20 (28.98%) had high NIHSS scores. In this subgroup, we examined caspase-3 together with other variables. The results showed that patients with exitus presented lower levels of caspase-3 (T1: 5.85 (3.15; 7.1) vs. 3.8 (3.15; 7.1) *p* = 0.027) (Figure 3). As expected, a higher NIHSS at 48 h was also associated with adverse outcomes (16 (15; 20) vs. 20 (15; 20); *p* = 0.001).

## 3. Discussion

The medical field of neurology and emergency medicine still has a pressing need for reliable biomarkers to ease the rapid diagnosis of AIS. Prompt identification of such patients is critical for the timely initiation of treatments, as a late intervention can lead to irreversible damage and worsened patient outcomes such as long-lasting disability or even death [17]. Caspase-3 is known as a pivotal mediator of cell death during the acute phase of stroke, and recent evidence points towards its involvement in other processes unrelated to neuronal death [18,19]. In this work, we aimed to explore the value of caspase-3 serum levels in differentiating AIS patients from control subjects enrolled consecutively in a Romanian hospital. Secondly, we sought to evaluate the association of this marker with the clinical and biological features of AIS.

To date, several studies have been conducted on other populations. Montaner et al. investigated the elevation of caspase-3 serum levels following cerebral ischemia and found that caspase-3 was the most predictive biomarker for the differential diagnosis of AIS [8]. Research on human brain tissue found that cleavage products of caspase-3 are associated with neural and neurovascular damage in conditions such as stroke, hemorrhage, and traumatic brain injury [20]. These markers have been detected in cerebrospinal fluid and blood, making them valuable in identifying pathways related to apoptosis and inflammation in brain conditions. Activation of caspase-3 induces neuronal and glial cell death, subsequently triggering pathological processes leading to chronic neurodegenerative diseases, such as dementia. Therefore, the accumulation of cleaved caspase-3 could serve as an early marker of neurodegenerative processes, and potentially become an important target for novel therapies [21,22]. Rosell et al. also showed a significant difference between AIS patients and controls. Moreover, they showed that the levels of caspase-3 increased proportionally with the infarct area on MRI acquisitions [14]. Lynch et al. measured serum levels of caspase-3 to diagnose AIS within the first 3 h of cerebral ischemia onset [23]. In our patients, caspase-3 levels were higher in AIS patients within the first 24 h from stroke onset and remained relatively constant even after 48 h. This suggests that caspase-3 could be used as an aid in the differential diagnosis of AIS from non-AIS patients during this timeframe, should the patients not present themselves to the hospital immediately.

In terms of the dynamic levels of caspase-3, according to Asahi et al. [24], there was a rise in caspase-3 mRNA levels in the rat brain one hour after focal ischemia. Namura et al. [25] found caspase-3 and its fragments in the mouse brain during reperfusion after a 2 h middle cerebral artery blockage. Qi et al. demonstrated that caspase-3 levels increased slowly after 8 h, showed a dramatic increase after 24 h, and began to decrease after 72 h [26]. Fink et al. demonstrated that caspase activation occurs up to 9 h after reperfusion. This suggests that an extended treatment window for caspase inhibition could be plausible after stroke [27]. This hypothesis has been explored in animal studies showing that pharmacological inhibition of caspase-3 can be beneficial in protecting against stroke [28,29,30]. These results were also supported by Wenying et al. showing that caspase-3 has a role in stroke recovery and its modulation could help promote the regeneration of neurons [18]. In our study, we measured caspase-3 levels at two timepoints and showed that the levels were higher in AIS patients at both 24 h and 48 h after hospital admission.

In terms of clinical features, Gafuroğullan et al. showed that there were no differences in caspase-3 levels related to patients’ sex, NIHSS scale values, or cerebral infarction volume, on 100 AIS patients with comorbidities similar to those in our cohort: diabetes, smoking, hypertension, coronary syndrome [31]. Moreover, they concluded that caspase-3 levels were not significant indicators of either the presence or absence of stroke or the duration of stroke onset. Rosell et al. [14] did not find significant associations between serum caspase-3 levels and risk factors other than atrial fibrillation. Similarly, Montaner et al. [8] determined that the serum level of caspase-3 was significantly higher in patients with atrial fibrillation and that the combination of serum caspase-3 levels and D-dimers could be used to differentiate between stroke and other neurological events mimicking stroke in the clinic. Linfante et al. did not find any correlations between the NIHSS score and the volume of the cerebral infarction, but their study considered only ischemia in the territory of the posterior cerebral circulation [32]. Conversely, Bustamante et al. compared the serum level of caspase-3 in patients with ischemic stroke, hemorrhagic stroke, and other conditions mimicking stroke and found no significant differences among them [33]. The differences obtained in the studies of various authors could be due to the timing of caspase-3 serum level measurement and therefore the time elapsed from the onset of AIS to presentation at the hospital. This could also explain why caspase-3 serum level correlates better with the volume of infarction determined on a second MRI evaluation performed 4.5 h after onset. In our study, caspase-3 serum levels were higher in AIS patients, making them more likely to receive an AIS diagnosis in the emergency room. This is of particular importance for AIS patients who do not present significant changes on their CT scans, rendering their cases difficult to diagnose. This finding has great potential, as markers for diagnosis of stroke in the absence of positive imaging could be of significant clinical relevance.

An interesting finding was that lower levels of caspase-3 were found in AIS patients presenting an early death, although these results should be taken with caution given the small number of cases in this subgroup. In a rat model of permanent and transient middle cerebral artery occlusion, caspase-3 remained significantly lower within 24 h of permanent middle cerebral artery occlusion. Reperfusion, on the other hand, was associated with a marked increase in caspase-3 activity [34]. Given these results, we hypothesize that patients who died early had a higher degree of occlusion, probably complete, which resulted in a poorer early prognosis. Thus, in these cases, reperfusion did not occur, preventing an increase in caspase levels in the serum.

The main limitations of our study are related to the relatively small number of patients and the single-center design. Moreover, the short follow-up period could potentially limit the results regarding the prognostic value of caspase-3 in AIS, which could be evaluated only in a longer timeframe.

The strengths of our study rely on the homogenous cohort of patients, and the repeated measurements of caspase-3 serum levels which showed the reliability and relatively constant values of this marker.

More research is needed to fully understand the significance of caspase-3 as a potential biomarker for AIS and its impact on neuronal death. It is important to investigate if blocking caspase-3 and its breakdown into different protein components could potentially help prevent neuronal apoptosis and enhance regeneration, given the conflicting results regarding the pathophysiological processes underlying AIS.

## 4. Materials and Methods

### 4.1. Study Design

This was an analytical, prospective, monocentric study approved by the Hospital Ethics Committee of “St. Apostle Andrei” Clinical Emergency County Hospital of Galati, Romania, with the reference number 524/07.01.2021. The study was conducted in accordance with the local ethical guidelines for research on humans and the 1964 Helsinki Declaration.

### 4.2. Patients’ Selection

The study included consecutive patients admitted at the Emergency Clinical Hospital of Galati-Clinical Neurological Department for clinical signs and symptoms of ischemic stroke in the previous 24 h before hospital presentation.

Inclusion criteria for the AIS group were as follows: clinical profile suggestive of AIS with an available brain CT scan performed within the first 24 h and a CT ASPECTS score of 10 (i.e., no visible signs of AIS modifications) [16]; at least 18 years old; and written informed consent from either the patient or a family member. Exclusion criteria were as follows: patients with other neurological conditions and/or hemorrhagic stroke; concomitant infectious diseases; acute myocardial infarction; recent traumatic events or surgeries; previous diagnosis of cancer; and severe organ insufficiency (renal/hepatic). The inclusion criteria for the control group were as follows: patients without cerebral ischemic pathologies presenting at the same hospital for other health-related issues. The exclusion criteria were the same as for the study group. All patients were enrolled between January 2022 and May 2023.

### 4.3. Patient Data Collection

Demographic data, detailed clinical data related to AIS, and medical history were obtained from each patient or their family members. The National Institutes of Health Stroke Scale (NIHSS) was applied by a neurologist to all patients, at their presentation in the hospital, after 48 h, and at the time of their discharge. The subtypes of AIS were defined in accordance with the Trial of Org 10,172 in the Acute Stroke Treatment (TOAST) classification system [35] through the evaluation of brain CT scans as previously described [36].

### 4.4. Blood Work and Caspase-3 Dosing

Blood samples were collected in BD-vacutainers serum tubes without anticoagulants upon patient presentation for routine analyses. Samples were frozen at −20 °C until analysis. Serum caspase-3 was determined at two timepoints: T1—first measurement within the first 24 h from hospital admission and T2—second measurement at 48 h from hospital admission. Measurements were completed using a sandwich ELISA assay according to the manufacturer’s recommendations (Cat.No.: E-EL-H0017, Elabscience, Houston, TX, USA). The concentrations were interpolated from a standard curve after measuring the optical density of the reaction at a wavelength of 450 nm. All measurements were performed by trained laboratory personnel who were blind to the patient’s diagnosis.

### 4.5. Statistical Analysis

Statistical analysis was executed in R version 4.3.2 using the RStudio IDE 2023.12.1+402 [37] (R Foundation for Statistical Computing, Vienna, Austria). Nominal variables were summarized using absolute frequencies and percentages. Continuous variables were summarized using the median and the 25th and 75th percentiles. The Shapiro–Wilk test was used to test for the normal distribution of the continuous variables. Significant differences between groups were tested using Chi-Square tests and *t*-tests or Mann–Whitney tests, as appropriate. Correlation analysis between continuous variables was completed using the Spearman test. Logistic regression was used to calculate the hazard ratio for AIS diagnosis while the threshold for discrimination was calculated with the Youden index. A *p*-value of ≤0.05 was used as a threshold for statistical significance.

## 5. Conclusions

Overall, caspase-3 levels were not directly indicative of AIS clinical features such as severity or prognosis when considering all NIHSS classes. However, in patients with moderate/severe NIHSS, low levels of Caspase-3 could possibly suggest the presence of a complete artery occlusion with a high risk of early death.

## Figures and Tables

**Figure 1 ijms-25-06772-f001:**
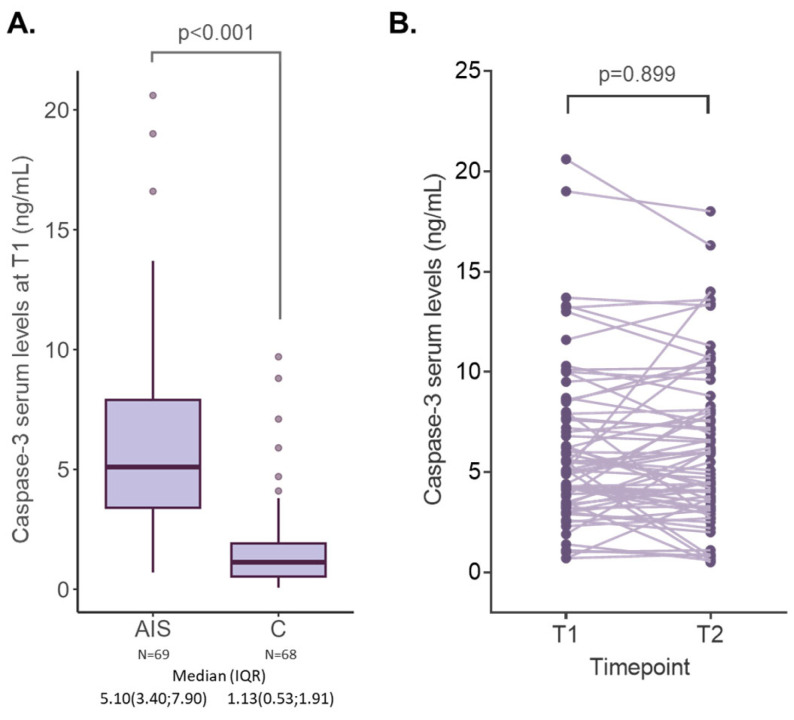
(**A**) Caspase-3 levels in AIS patients and control subjects. (**B**) Caspase-3 levels in AIS patients at the two timepoints (T1—within 24 h from hospital admission and T2—after 48 h from admission). AIS: acute ischemic stroke; C: control group; IQR: interquartile range.

**Figure 2 ijms-25-06772-f002:**
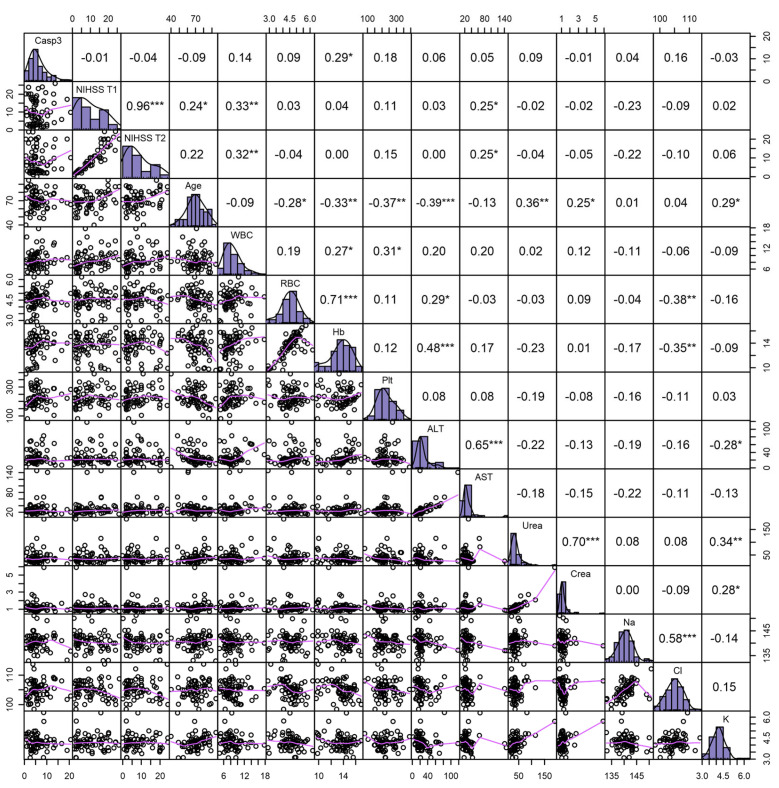
Correlation analysis between paraclinical investigations and caspase-3 serum levels. The values in the upper right quadrant of the graph represent the correlation coefficients, while the significance level is reported with symbols as follows: ***: 0–0.001; **: 0.001–0.01; *: 0.01–0.05; •: 0.05–0.10. The distribution histograms (Medium Purple) and density plots of the variables are presented on the diagonal while the lower left quadrant shows the individual data points with the model fit (Medium Orchid). Casp-3: caspase-3; WBC: white blood count; RBC: red blood count; Hb: hemoglobin; Plt: platelets; ALT: alanine aminotransferase; AST: aspartate aminotransferase; Crea: serum creatinine; NA: sodium; Cl: chloride; K: potassium.

**Figure 3 ijms-25-06772-f003:**
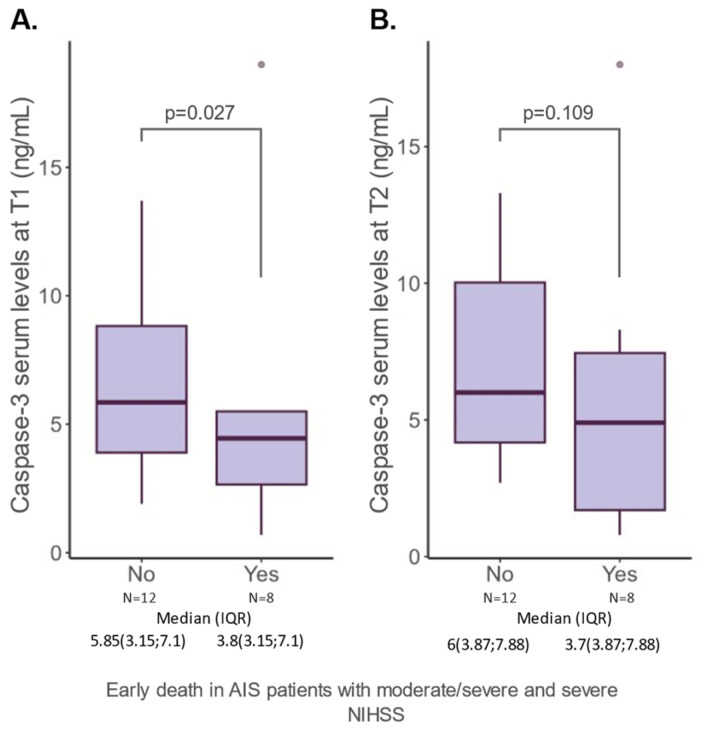
Serum caspase-3 levels at 24 h (**A**) and 48 h (**B**) from hospitalization in patients with AIS and NIHSS above 16 points. The datum depicted above the *p*-value was identified as an outlier (according to the Dixon test; *p* < 0.001) and subsequently excluded from the analysis but retained on the plot for data integrity and transparency. AIS: acute ischemic stroke; IQR: interquartile range; N: number of cases.

**Table 1 ijms-25-06772-t001:** Demographic characteristics and comorbidities of the AIS patients and controls included in this study.

Variable		Total N = 137	Control Group N = 68	AIS Group N = 69	*p*-Value
** Gender ** N (%)	**Female**	56 (40.88)	30 (44.12)	26 (37.68)	0.444
	**Male**	81 (59.12)	38 (55.88)	43 (62.32)	
** Age ** [years] mean ± sd		69.77 ± 10.69	69.28 ± 10.5	70.23 ± 11.01	0.605
** Residential background ** N (%)	**Urban**	81 (59.12)	36 (52.94)	45 (65.22)	0.144
	**Rural**	56 (40.88)	32 (47.06)	24 (34.78)	
** * Comorbidities * **					
** Atrial Fibrillation ** N (%)	**No**	94 (68.61)	51 (75)	43 (62.32)	0.110
	**Yes**	43 (31.39)	17 (25)	26 (37.68)	
** Dyslipidemia ** N (%)	**No**	60 (43.8)	34 (50)	26 (37.68)	0.146
	**Yes**	77 (56.2)	34 (50)	43 (62.32)	
** Diabetes ** N (%)	**No**	96 (70.07)	41 (60.29)	55 (79.71)	0.013
	**Yes**	41 (29.93)	27 (39.71)	14 (20.29)	
** Blood hypertension ** N (%)	**grade 1**	16 (11.68)	11 (16.18)	5 (7.25)	0.033
	**grade 2**	56 (40.88)	32 (47.06)	24 (34.78)	
	**grade 3**	65 (47.45)	25 (36.76)	40 (57.97)	

N: number of cases; sd: standard deviation; AIS: Acute ischemic stroke.

**Table 2 ijms-25-06772-t002:** Serum Caspase-3 levels and comorbidities and clinical characteristics of AIS patients.

Variable		Caspase-3 T1 Median (IQR)	*p*-Value
** * Comorbidities * **			
Atrial fibrillation N (%)	no	5.5 (3.25; 7.85)	0.827
	yes	4.95 (3.58; 7.73)	
Blood hypertension grade N (%)	1	3.3 (1.4; 6)	0.261
	2	5.2 (4.12; 7.18)	
	3	5.1 (3.48; 8.53)	
Dyslipidemia N (%)	no	5.3 (2.72; 9.3)	0.75
	yes	5.1 (3.7; 6.55)	
Obesity N (%)	no	5 (3.38; 7.78)	0.269
	yes	6.2 (5.5; 7.9)	
Diabetes mellitus N (%)	no	5.1 (3.35; 8.25)	0.581
	yes	4.7 (4.15; 5.8)	
** * AIS features * **			
AIS type N (%)	Cardioembolic	4.95 (3.58; 7.73)	0.827
	Atherothrombotic	5.5 (3.25; 7.85)	
Ischemic territory (anatomic segment) N (%)	Left sylvian	5.5 (3.4; 8.75)	0.489
	Right sylvian	5.05 (3.45; 7.05)	
	Vertebrobasilar	5.75 (4.3; 8.28)	
NIHSS severity at admission N (%)	0–4. minor stroke	5.55 (3.6; 7.45)	0.543
	5–15. moderate stroke	5 (3.45; 7.8)	
	16–20. moderate to severe	4.9 (3.15; 5.5)	
	21–42. severe	8 (6.2; 9.5)	
NIHSS improvement at 48 h N (%)	no	5 (3.15; 6.95)	0.581
	yes	5.55 (3.67; 8.45)	
Early death N (%)	no	5.1 (3.5; 8)	0.735
	yes	4.45 (2.65; 5.5)	

## Data Availability

Due to ethical restrictions, data presented in this manuscript may be provided by authors upon reasonable request.

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
