# Peer review of "Serum Caspase-3 Levels as a Predictive Molecular Biomarker for Acute Ischemic Stroke"

_ijms, 2024, doi:10.3390/ijms25126772_

Round 1

Reviewer 1 Report

Comments and Suggestions for Authors

The manuscript entitled „Serum Caspase-3 Levels as a predictive molecular biomarker for Acute Ischemic Stroke “  is an interesting prospective study evaluating the serum levels of Caspase-3 in patients with acute ischemic stroke 18 (AIS) and in control patients without ischemic event.

However, significant changes should be made to improve the manuscript. 

The part of the text needs to be significantly improved – the authors used wordy sentences and incorrect phrases.

Comments:

1. Caspase 3 is written differently in the text, some in capital letters and some in small letters - this should be harmonized

2. Small number of AIS patients 

3. The authors did not explain the selection of control patients and the differences between the AIS patients

4. Introduction: in review lines 35-37: missing reference. In more detail (lines 46-47), the reference should have been placed under number 1.

5. Review: lines 46-49 are written unclearly, so they should be revised to make them more understandable

6. The word "centres" is misspelled

7. The expression (line 58): "such as some" is incorrect as an expression

8. Physical values ​​for caspases are not given in the tables and text

9. The vascular territory listed in Table 2 is uncommon - it does not represent the territory of a single blood vessel: the usual terms related to the territories from the areas ACM, ACI, ACP, then the areas of large and small blood vessels L VO and SVO should be listed

10. Figure 2 is written twice - on the 6th page it should be Figure 3

11. In Material and Methods: the authors did not explain the way of diagnosing the infarction: did they use only native CT, CT perfusion, or CT angiography? Did they use MRI for diagnosing acute infarction there – it is especially important in diagnosing the ischemic lesions in the posterior circulation. How was the occlusion diagnosed? How many vessels were occluded? Did they evaluate the correlation of the number and type of occlusion with the serum values of caspase-3?

12. The Cut off value for caspase 3, which could mean the threshold for pathologically elevated caspase 3, is not specified

13. The references are incorrectly written  (ref. 25, ref .29)

Comments on the Quality of English Language

The part of the text needs to be significantly improved – the authors used wordy sentences and incorrect phrases.

Author Response

please find attached out point by point responses

Reviewer 2 Report

Comments and Suggestions for Authors

Minor comments are listed below

1. Could the authors highlight what are novel aspects in their work in terms of the relationship between apoptosis and stroke?
2. Are there any clinical consequences of the results obtained?
3. Spaces are omitted in many places.
4. Have the necessary assumptions for the use of the parametric t test been checked?

Author Response

(The authors gave the same response as above.)

Round 2

Reviewer 1 Report

Comments and Suggestions for Authors

The manuscript entitled „Serum Caspase-3 Levels as a predictive molecular biomarker for Acute Ischemic Stroke “ was thoroughly revised by the authors based on the feedback provided by the reviewers. The authors calculated the power of our study and added this information to the manuscript. They also added information regarding the inclusion and exclusion criteria of the control patients and explained how to diagnose the infarction.

Due to the improved writing, the manuscript is now suitable for publication and should be considered for acceptance.